



# Hydrological Drought across Peninsular Malaysia: Implication of drought index

Hasrul Hazman Hasan[1], Siti Fatin Mohd Razali[1], Nur Shazwani Muhammad[1], Asmadi Ahmad[2]

[1]Department of Civil Engineering, Faculty of Engineering & Built Environment, Universiti Kebangsaan Malaysia, 43600
UKM Bangi, Selangor, Malaysia
[2]Office of Klang River Basin, Department of Irrigation & Drainage Kuala Lumpur, Kolam Takungan Batu, Batu 4 ½ Off
Jalan Ipoh, 51200 Kuala Lumpur, Malaysia

*Correspondence to*: Siti Fatin Mohd Razali (fatinrazali@ukm.edu.my)

**Abstract.** Drought is considered a damaging natural disaster for economic, societal, and ecological impacts. The challenge
of drought is to determine the drought characteristics, frequency, duration and severity, vital for drought's impact control and
mitigation strategies. This paper adopts the spatial pattern of Streamflow Drought Index (SDI) for three, six, nine and twelve
months for the tropical climate at Peninsular Malaysia. About 40 years of daily streamflow data based on 42 hydrological
discharge stations were analyzed to obtain these indices. The area under drought stress during the study period at different
time scales is stable and approximately 24% of the total area. The years 1997-1999, 2002 and 2016-2018 mark the most
critical drought years, with more than 48% of the entire basin area under hydrological drought. According to the spatial
evaluation of drought characteristics, short-term droughts are frequent in most regions, with relatively high severity and
frequency in Northeast and Southeast of Peninsular Malaysia, where the maximum frequency reached 35.7% and 42.8%,
respectively. This outcome emphasizes the importance and necessity of the basin's drought action strategies. Early detection
of a probable hydrological drought can improve in the implementation of drought prevention or mitigation strategies.

## 1 Introduction

Drought may be considered a common phenomenon resulting from climate changes and can be worsened by anthropogenic
influences. Recent years have focused on the global drought scenario caused by increased water demand and climate change.
Droughts in the region differ from other natural disasters in slow onset and prolonged duration, which can last up to months
or years (Yang et al., 2017). All the world's continents experience the consequences of drought conditions. Droughts are
anticipated to become more frequent, intense, and prolonged in diverse places globally throughout time (Leta et al., 2018).
As these characteristics increase, the world percentage subjected to extreme drought is expected to rise, from 1% to 30% in
the 21st century. This indicates that the number of severe drought occurrences and the duration they last are expected to
increase. (Siderius et al., 2018).



Droughts may have a significant impact on a broad variety of socio-economic activities, including farming, water supplies, wildlife, aquaculture, tourism, and transportation, as well as hydroelectric power generation. Due to the frequency and duration of drought, it is necessary to consider developing a comprehensive drought monitoring system capable of providing early warning of the beginning and end of droughts. Drought severity, duration, frequency and spatial extent are the most valuable indicators for assessing the spatio-temporal characteristics of droughts in a given region, which is considered one of

the essential aspects of drought disaster mitigation (Hasan et al., 2019). Drought-related economic, social, and environmental impacts might be minimized using this information.

Drought is a natural feature of the climate usually associated with dry and warm weather over an extended period causing less than the average volume of water available at the land surface. The development of a drought is a slow process, and it is

often hard to detect early detection. This difficulty is determining when a drought begins and, conversely, when it ends. Drought is a complex phenomenon that can be classified as meteorological, hydrological, agricultural, and socio-economic (Barella-Ortiz and Quintana-Seguí, 2019). Briefly, a meteorological drought resulted from the absence or lack of precipitation compared to the average precipitation over a prolonged period; an agricultural drought developed due to the unavailability of water at different stages of crop development to sustain production; a hydrological drought forms when

water supply in storage systems, such as aquifers, reservoirs, and rivers, falls below the level required for their operational and ecological uses; and a socio-economic drought sets in when the shortage of water adversely affects the supply of an economic good or service or has an impact in the broader economy (Choi et al., 2018; Li et al., 2020; Wang et al., 2016). Among these classifications, assessment of hydrological drought has more importance in the water resources management perspective.


Hydrological drought is related to events that caused inadequate water supply from streams, reservoirs and aquifers under a given water resources management system (Barker et al., 2016), including streamflow drought and groundwater drought. Although precipitation deficiency is the main reason for hydrological drought, it takes longer to manifest a deficiency in the river flow or the groundwater level. Hydrological drought is best characterised by hydrological parameters such as

streamflow. An understanding of this parameter is necessary to develop mitigating measures for the impacts of droughts. The most straightforward method of monitoring drought conditions is through the use of a drought index. The index is presented as a quantitative method for identifying when drought occurrences begin and finish, thus minimizing drought severity (Ward, 2013). Drought indices enable identifying drought characteristics such as timing, length, intensity, and spatial distribution, thereby assisting decision-makers in managing drought mitigation strategies.


Several hydrological drought indices have been established throughout the years to evaluate hydrological drought. Indices developed under the hydrological approach heavily rely on streamflow, runoff, reservoir, and groundwater levels data. This approach aims to quantify the delayed effects of drought in the hydrological system, hence reflecting a deficiency of water


resources over some time. However, several indices are using only streamflow data, namely, Regional Streamflow Deficiency Index (RSDI), Standardized Streamflow Index (SSFI), Streamflow Drought Index (SDI), Baseflow Index (BFI) and Regional Drought Area Index (RDAI) (Niemeyer, 2008).

In recent years, several researchers have applied the Streamflow Drought Index (SDI) methodology for streamflow drought investigations (Akbari et al., 2015; Barker et al., 2016; Hong et al., 2015; Tigkas et al., 2012). SDI developed a drought
analysis using continuous streamflow values. This index was used to evaluate several countries' hydrological droughts, including the United States, India, Iraq, and Iran. Tabari et al. (2013) used SDI in West Azerbaijan, Northwest of Iran, with 39487 km$^2$ for 34 years of study (Tabari et al., 2013). Four overlapping periods were utilised within each hydrological year. According to the study, practically all stations suffered extreme drought episodes, mainly in the recent 12 years. During the examined period, Iran experienced the driest hydrological years from 1998 to 2001. A study was conducted by Manikandan
& Tamilmani (2015) using SDI and theory of runs (ToR) assessments in the Aliyar sub-basin, India, for determining the hydrological drought characteristics namely the drought duration, severity, intensity and frequency (Manikandan and Tamilmani, 2015). The drought severity-frequency curve was drawn based on the weighted annual cumulative drought severity and average annual drought severity. The findings from this study showed that the persistent and prolonged drought of 2002 to 2004 was significantly affecting the study area. These studies have proved the ability of SDI in determining the
hydrological drought at spatial and temporal scales.

Recently, climate change has prompted severe consequences to the entire world, such as extraordinary flood, extreme drought, snowfall, heatwave, and natural disaster. For instance, the El Nino event arises from the increase in seawater surface temperature, and it has a cumulative effect on global warming. In the record, the El Nino event was occurred and
brought critical damage across the globe in the year 1997-1998 (Frappart et al., 2018). For example, Indonesia has experienced a drought event; meanwhile, excessive rainfall resulted in the mudslide in the United States, which cause casualties (Yasa et al., 2018). During the El Nino event, the temperature in Malaysia has increased, and the forest fires took place in Malaysia and contributed to serious haze circumstance in 1997-1998. During this time, precipitation has decreased, resulting in more extreme dry seasons in Malaysia (Bong and Richard, 2019; Tan et al., 2019).

El Nino reappeared in 2016, causing hydrological droughts in Malaysia. This resulted less than half of the storage in the reserve water level in seven dams: Timah Tasoh (Perlis), Bukit Kwong (Kelantan), Beris Padang Saga, Muda (Kedah), Bukit Merak (Perak) , and Labong (Johor) (Tan et al., 2019). Hydrological drought results in a water crisis and, therefore, the production of crop reduced. Thus, in order to minimise the impact of climate change on water supply, it is required to
conduct effective hydrological analysis and management. Therefore, in this study, SDI and ToR were employed in the long-term distribution analysis of hydrological drought events at 42 stations in Peninsular Malaysia.



Since the drought has a massive influence, the drought indices should be used as an indicator to inspect the characteristic of an occurrence in a drought. Therefore, drought indices are essential to make a strategic decision on water management during the drought happen. Previous researchers, e.g. Zin et al. (2014), Khan et al. (2017) and Yusof et al. (2014) studied the meteorological and agricultural droughts in Peninsular Malaysia (Khan et al., 2017; Yusof et al., 2014; Zin et al., 2014). In contrast, this paper focused on hydrological drought, which uses long term streamflow data. The streamflow is the most crucial part of the hydrological cycle. Historical drought analysis provides information for effective drought monitoring in the future. Researchers have adopted several procedures to characterise droughts. A technique widely used in the study of droughts is the use of drought indices.

In comparison, the Streamflow Drought Index (SDI) is a convenient and straightforward index for evaluating hydrological drought. Due to the scarcity of research on hydrological drought monitoring using SDI, this study focused on assessing hydrological drought using SDI for historical streamflow data. Thus, this study aims to investigate the hydrological drought in Peninsular Malaysia from 1978 to 2018 (40 years) using Streamflow Drought Index (SDI) as the indicator, with the following objectives: (i) to identify and characterise hydrological drought using the ToR; and (ii) to determine the temporal and spatial patterns of hydrological drought based on the short- and long term durations. The findings of this study can be utilised to evaluate water resources regionally and prepare for drought management. Planning authorities need to identify temporal and spatial patterns of established droughts, access aridity prognosis, and publicly communicate the results for a better mitigation plan and preparedness.

## 2 Methodology

The hydrological droughts in Peninsular Malaysia were determined using the SDI based on 40 years of streamflow data. These data were then analysed in four significant steps. The first step was the analysis of SDI using monthly streamflow data. In the second step, the frequency of SDI for each classification was evaluated. Then, each station's identified hydrological drought characteristics were further analysed using the theory of runs (ToR). Finally, the Inverse Distance Weighting (IDW) method was applied to perform the temporal and spatial analysis on hydrological drought characteristics across Peninsular Malaysia.

### 2.1 Study area

Peninsular Malaysia is located between 1° to 6° N latitude and between 100° to 103° E longitude (Mamun et al., 2010). Peninsular Malaysia is a tropical country with two distinct rainy seasons, namely Southwest and Northeast Monsoons. From June to September, the Southwest Monsoon, often known as the summer monsoon, occurs, whereas the Northeast Monsoon occurs from November to March. This annual monsoon results in raining and dry events in Malaysia (Yusof et al., 2014).



Southwest Monsoon that originates from the deserts of Australia exert a dry monsoon season to Northern Peninsular
Malaysia with lesser rainfall because of the blockage by the high mountain ranges in Sumatra. While in Northeast Monsoon,
which originates from China, streamflow along the North Pacific tends to bring more rainfall to Malaysia.

Peninsular Malaysia consists of 12 states which is Johor, Melaka, Negeri Sembilan, Wilayah Persekutuan, Selangor, Pahang,
Perak, Kelantan, Terengganu, Pulau Pinang, Kedah and Perlis. In this study area, there are 42 streamflow stations with 40
years of continuous historical streamflow data, as shown in Figure 1.

**Figure 1: The location of streamflow stations in Peninsular Malaysia.**

For the study, the monthly streamflow data were collected from the Department of Irrigation and Drainage (DID), Malaysia,
from 1978 to 2018. As aforementioned, there are 42 well-functioning streamflow stations for Peninsular Malaysia, where the
details of each station are shown in Table 1. About 17 out of 42 stations have less than 40 years of streamflow data as their
installation date are lesser than 40 years.
**Table 1 Information on Streamflow Stations in Peninsular Malaysia**

**2.2 Streamflow Drought Index (SDI)**

The Streamflow Drought Index (SDI) proposed by Nalbantis & Tsakiris (2009) was used to assess the extent of hydrological
drought in Peninsular Malaysia at different time scales (Nalbantis and Tsakiris, 2009). Streamflow data is the only crucial
variable in generating SDI. The main advantage of SDI is that it requires fewer data than other indices, such as the Palmer
Hydrological Drought Index, which need streamflow and rainfall data. The selection of SDI is because of the availability of
streamflow data. As the SDI index relies on streamflow data, it is unsuitable for watersheds analyses with no streamflow
station. Streamflow reconstruction is usually performed for streams with partial streamflow data (Hasan et al., 2021).

The SDI at varying intervals time scales indicates droughts of varying duration within a region. For instance, the three-month
drought index (SDI-3) reflects seasonal water condition, providing vital information for irrigation in agricultural sectors
(Akbari et al., 2015). The six-month drought index (SDI-6) can depict drought conditions over a half-year period (Wambua,
2019). The drought index for 12 months (SDI-12) is excellent for assessing the effect of climate change on regional water
supplies (Tabari et al., 2013).

For a relatively more detailed drought index, the SDI can be computed based on the monthly streamflow value (Sardou and
Bahremand, 2014). It is utilised in identifying hydrological drought at different time scales and thus offers the advantage of
controlling hydrological drought and water supply in the short, medium and long term. The time series of monthly



streamflow volumes ($Q_{i,j}$) are consecutive and accumulated based on their time duration (k) within the hydrological year. The streamflow volumes can be calculated as Eq. (1):

$$V_{i,k} = \sum_{j=1}^{3k} Q_{i,j} \, , \tag{1}$$

where $Q_{i,j}$ denotes monthly streamflow volumes, i denotes the hydrological year, j refers to the month of that year, k denotes the period length and $V_{i,k}$ is the cumulative streamflow volume for an i-th hydrological year with a period duration of k. i = 1, 2, ……., 40 (hydrological year); j = 1, 2, ….., 12 (where j = 1 for January and j = 12 for December); k = 1, 2, 3, 4 (k = 1 for January to March, k = 2 for January to June, k = 3 for January to September and k = 4 for January to December). Next, the long term mean ($\overline{V}_k$) and standard deviation ($s_k$) of the cumulative streamflow volume ($V_{i,k}$) is calculated to define the SDI for the k-th reference period within the i-th hydrological year, as follows in Eq. (2):

$$SDI_{i,k} = \frac{V_{i,k} - \overline{V}_k}{s_k} \, , \tag{2}$$

where k = 1, 2, 3, 4; and i = 1, 2, 3, …., N.  The categories of hydrological drought are determined based on the computed SDI. The SDI values have been divided into categories ranging from extremely wet to extremely drought (Nalbantis and Tsakiris, 2009). Wet conditions are defined as values larger than 0, whilst drought conditions are defined as values less than 0. Table 2 provides descriptions of Streamflow Drought Index (SDI) values.

**Table 2 Hydrological Drought Classification by SDI**

### 2.3 Estimation of frequency of SDI

The frequency of each category of SDI is estimated using the following Eq. (3):

$$F_{m,k} = \frac{n_{m,k}}{N} \, , \tag{3}$$

Where $F_{m,k}$ is the frequency of each category m in reference period k (where k = 1, 2, 3, 4); N is the total number of sample years available and $n_{m,k}$ is the number of drought occurrences for each category m within the k reference periods.

This study classified the hydrological drought frequency into five classes, which are rare (0% – 20%), less (20% – 40%), often (40% – 60%), frequent (60% – 80%), and extremely frequent (80% – 100%) (Hong et al., 2015), for understanding and interpreting the spatial-temporal of hydrological drought in Peninsular Malaysia.





## 2.4 Identification of drought events

It is hard to identify the onset and the end of a drought event in defining drought characteristics. However, with the drought

index, one can monitor the drought monitoring and analyse the drought characteristics. Once the Streamflow Drought Index is calculated based on the data series of streamflow, it is necessary to take specific criteria to detect the drought events. This study applied the ToR, which initially proposed by Yevjevich, for determining the hydrological drought characteristics (Razmkhah, 2017). Severity, duration and intensity are three crucial components for drought characteristics (Figure 2). Following Nalbantis & Tsakiris (2009), the successive sequence of months with SDI values ($X_t$) below the threshold value

($X_{-1}$) is defined as a hydrological drought event (Nalbantis and Tsakiris, 2009).

**Figure 2: The three components of a drought event for drought characteristics.**

A drought event can be defined as when the index is smaller than -1.0 in at least one period, as shown in Table 2. Otherwise,

it will be neglected since it is only a minor drought event (Hong et al., 2015). The drought initiation period is the month in which the SDI values fall below -1.0, indicating the start of a drought episode. The drought duration is the period between the time of occurrence of drought and the time of completion. The accumulated drought index defines the severity of the drought during a drought event. The drought intensity is measured as the drought severity divided by the duration, and the drought peak is the highest negative SDI value during the drought event. The drought onset was established at the beginning

of the period when SDI was negative for an extended period. The drought was expected to cease in the first month for the SDI to turn positive.

Drought severity ($S_{SDI}$) was calculated during $n$ months drought event as the sum of absolute SDI values throughout that duration, as indicated in Eq. (4). Moreover, the drought intensity ($I_{SDI}$) was calculated as the mean $S_{SDI}$ over the drought event in Eq. (5):

$$S_{SDI} = \sum_{j=1}^{n} |SDI_j| \,, \tag{4}$$

$$I_{SDI} = \frac{S_{SDI}}{n} \,, \tag{5}$$

## 2.5 Hydrological drought zoning

By minimising the distance effect of its sample site, the Inverse Distance Weighting (IDW) evaluates the neighbourhoods of

the selected sites where the variable is shown (Jahangir and Yarahmadi, 2020). As demonstrated in Eq. (6), the Inverse Distance Weighting (IDW) approach was employed in this study to interpolate and analyse the spatial variations of hydrological drought characteristics across Peninsular Malaysia.



$$\hat{Z} = \frac{\sum_{i=1}^{n} \frac{Z_i}{d_i^k}}{\sum_{i=1}^{n} \frac{1}{d_i^k}}, \tag{6}$$

$\hat{Z}$ is the estimated value at an unsampled point, n is the number of control point used for the estimate, k is the power of which

distance is raised, d is the distance from each control points to an unsampled point.

Peninsular Malaysia's hydrological drought characteristics were spatially interpolated to visualise the SDI's spatial variability. To verify the accuracy and correlation of the interpolated values with the originally recorded data in the region, the IDW interpolation method was cross-validated with the standard statistical tests, including the Root Mean Squared Error

(RMSE) and the coefficient of correlation (R) (Aghelpour and Varshavian, 2020). The SDI index was used for zoning drought, which was calculated over 12 months. Then, the station names, Universal Transverse Mercator (UTM) coordinates, and SDI data were entered into the Arc Map software. The IDW approach was then used to execute drought zoning.

## 3 Result and discussion

### 3.1 SDI analysis

The analysis results are compared to the standard drought criterion in Table 2 to assess the hydrological drought. Figure 3 depicts the SDI-12 over 40 years. Positive numbers indicate wet conditions, whereas negative values indicate drought.

**Figure 3: Colour-coded table of SDI-12 for 40-year time series**


In Figure 3, all streamflow stations experienced at least one severe drought in 40 years (-2.0 ≤ SDI < -1.5). After 2010, about 22 stations experienced severe and extreme droughts for SDI-12 (January-December). The most severe drought can be identified at station S28, located along the Perak River in 2012 - 2018. Furthermore, all the extreme drought events occurred in the last nine years from 2012. In general, the years of 1996 - 1998 with four stations in Selangor (S14, S17, S20 and S21)

and one station in Pahang (S18), 2002 - 2003 with a total of five stations (Three stations in Selangor – S11, S13, S20; one station in Perak – S25; and one station in Kedah – S33) and 2012 - 2018 with 15 stations were the driest. The 15 stations at Johor (S01, S02 and S06), Melaka (S05), Negeri Sembilan (S10 and S12), Selangor (S21), Wilayah Persekutuan (S15 and S16), Pahang (S18), Terengganu (S23 and S35), Perak (S27 and S28) and Perlis (S40). Some stations were repeated occurred extreme hydrological drought during the period time studied (40 years) are S21 (Selangor), S18 (Pahang) and S28 (Perak). In

other words, the most severe hydrological drought occurred during the hydrological years of 1996 - 1998, 2002 - 2003 and 2012 - 2018. The most severe hydrological drought is occurred at station S21, located along the Bernam River, Selangor, in 1997-1998 and 2014-2016. According to Shaaban and Low (2003), Peninsular Malaysia was severely affected by the





1997/1998 El-Nino drought event, particularly in Selangor districts and the Central region, and the effect lasted until
September 1998 (Shaaban and Low, 2003). When comparing the 1997/1998 El-Nino to the 2015/2016 El-Nino drought,
Fung et al. (2020) discovered that high temperatures impacted the dry months in these times (Fung et al., 2020).

The SDI results in the study region show that moderate and severe drought occurrences likely to follow one another for more
than a year on average. For example, moderate drought was experienced in 2012 and 2013. In the following year, 2014 and
2015, severe drought conditions were observed. In the successive years, 2015 to 2018, extreme drought conditions were
recorded. The results from each station in Peninsular Malaysia were characterised into drought categories, events, and
frequency. Then, visual interpretations for each region were created by mapping the characteristics into a raster surface
spatial map to analyse the variation over space (Figure 4 to 6).

### 3.2 Spatial and temporal SDI analysis

Drought zonings were also evaluated since most streamflow stations experienced prolonged droughts with more than two
drought episodes in a row throughout the study period. The drought zoning of the study region was investigated using IDW
over periods of 3, 6, 9, and 12 months. Figure 4 illustrates the spatial and temporal zonings for SDI-3, SD-6, SDI-9, and
SDI-12 over 10 years.

**Figure 4: Spatial and temporal distribution of SDI (a) three and six months and (b) nine and twelve months for ten
years interval**

A three-month SDI represents short-term hydrological drought and delivers seasonal streamflow estimations. The 6- and 9-
month SDIs reflect medium-term trends in streamflow and can be used to compare streamflow between seasons. A six-
month SDI may also begin to be correlated to anomalous streamflow and storage reservoir situations. The SDI reflects long-
term streamflow patterns in 12 months. Referring to Figure 4, the 1980s-1990s and 2010s are the years when more than half
of the basin's entire area is affected by drought (SDI < -1). Drought severity was calculated at ten-year intervals in the 1980s,
1990s, 2000s, and 2010s to see any changes across the region over time. Drought severity increased throughout time, with
values of 1.05 changing to 10.50 (SDI-3), 1.45 to 23.80 (SDI-6), 3.32 to 25.12 (SDI-9) and 3.74 to 28.90 (SDI-12).


Based on the time scale of SDI-3, the moderate hydrological drought experienced in the 1980s with a large drought area are
28.57%, followed by severe hydrological drought 4.76% area. In the 1990s, the hydrological drought of the river basin in
Peninsular Malaysia was generally severe, with a moderate drought area of 44.64%. Moreover, the hydrological drought
areas of SDI-3 accounted for 42.86% of the total area. For the time scale of SDI-6, the moderate hydrological drought is
decreased to 23.81%, based on drought area. The severe drought also decreased to 2.38% of the total area in the 1980s-





1990s. The pattern is the same as SDI-9. However, for SDI-12, the moderate hydrological drought is the same percentage of drought area with 42.86% of the total area. SDI-12 has not indicated any severe or extreme hydrological drought during the 1980s to 1990s. No extreme hydrological drought during the 1990s.

In the 2000s, the entire water basin of Peninsular Malaysia did not experience hydrological drought, except in the Northern region. The moderate hydrological drought areas of SDI-3, SDI-6, SDI-9, and SDI-12 accounted for 33.33%, 35.71%, 33.33%, and 26.19% of the total area. The most crucial drought years in the entire river basin occurred in the past decade (the 2010s), with more than 75% of the overall basin in Peninsular Malaysia experiencing drought, which coincided with the El Nino event. The year 2016 was the worst, with drought affecting 85% of the whole river basin, followed by drought

affecting 80% of the river basin's total in 1997 and 1998. For the different time scales, there are 2559 hydrological drought events for each drought category. For SDI-3, 505, 265 and 152 of drought event for moderate, severe and extreme drought, respectively. The total number of drought event for SDI-6 is 351 (moderate), 196 (severe) and 105 (extreme). The time scales of 9 months, 291 drought events (moderate), 171 drought events (severe) and 83 drought events (extreme). Lastly, for SDI-12, there are about 231, 133 and 76 drought events for moderate, severe and extreme drought, respectively. The area

under drought stress during the study period at different time scales is stable and approximately 24% of the total area with the increase in the time scale.

In the 1980s, based on SDI-3 to SDI-12, the spatial distribution of hydrological drought was changed from South-East to North-East of Peninsular Malaysia in general. However, in the 1990s, the spatial distribution mainly shows the East and

West Coast of Peninsular Malaysia. The spatial distribution of hydrological drought in the 2000s mainly focuses on the Northwest of the Northern region of Peninsular Malaysia. In the 2010s, the spatial distribution of hydrological drought was relatively complex. The spatial pattern was noticed distributed change from the year 2000. SDI-3 shows Northeast distribution; SDI-6 shows a specific Southeast distribution; SDI-9 shows a scattered distribution on the Northwest, and SDI-12 shows Southwest distribution areas.


The moderate level of hydrological drought be inclined to the Northeast and mainly distributed on the west coast of Peninsular Malaysia. The spatial distributions of the 1980s to 2010s are mainly distributed in the Southeast and Northwest of Peninsular Malaysia at a severe level. For the extreme level of hydrological drought, the spatial distribution of the 1990s in Northwest distributed and 2010s in Southeast and Northeast distributed.  Thus, the drought area of the river basin in

Peninsular Malaysia expands with the increase of time scales. The spatial and temporal SDI analysis revealed that the SDI-3 and SDI-6 might be misleading in the regions normally dry for six months. The SDI-3 may be used to determine when the dry season begins and ends. However, having a drought index for more extended periods is essential. For example, three months may occur in the middle of a long-term drought, but this would only be noticeable over more extended periods, such as 12 months.





### 3.3 Hydrological drought events

Drought indices simplify the complicated relationships involving hydrometeorological parameters, allowing them to identify drought propagation characteristics effectively. Combining the obtained SDI and the ToR, the drought events and important characteristic variables can be identified. As shown in Fig. 2, a drought event is a series of events in which the SDI value falls below the given threshold level, which in this case is set at -1. Combined with the monthly sequence data of SDI between 1978 to 2018 in Peninsular Malaysia, the threshold levels $R_1$ = -1 and $R_2$ = -2 were selected to identify and extract drought events. The hydrological drought characteristics of drought events have been evaluated for the reference period (SDI-3, SDI-6, SDI-9, SDI-12) and shown in Figure 5. Almost all stations exhibit an astounding number of drought events. It has also been discovered that during the first two reference periods (SDI-3 and SDI-6), every station experienced more drought events. The findings demonstrate that nearly every station has the same number of drought pattern. Based on the ToR and SDI, 391, 439, 408 and 392 drought events were identified in Peninsular Malaysia between 1978-2018 for SDI-3, SDI-6, SDI-9 and SDI-12, respectively.

**Figure 5: Spatial and temporal of hydrological drought events (a) three and six months and (b) nine and twelve months in Peninsular Malaysia**

The hydrological drought of all historical drought events (the 1980s, 1990s, 2000s and 2010s) with different durations are shown in Figure 5. The analysis found that hydrological drought was mainly distributed in the central and east coast region of Peninsular Malaysia during the early decades of the study period. Due to the study's location in a humid tropical climate, it is also assumed that groundwater replenishment occurred during the early stages of drought events. The shorter period of the event can be explained by referring to results suggesting that the primary cause of the event was the higher demand for water during the dry spell (Mussá et al., 2015).

In year 1980s, based on the spatial distribution (SDI-3 and SDI-6), the hydrological drought events in Peninsular Malaysia in the recent 40 years showed a high number in the Northeast and Southwest. For SDI-9 and SDI-12, the spatial pattern shows the same trends, with the maximum number of hydrological droughts is 5 to 6 events in 10 years. In the 1990s, hydrological drought events declined compared to the total number of events recorded in the 1980s. SDI-3 and SDI-6 show that the maximum number of drought events is 7 to 8, mainly in the Northwest of Peninsular Malaysia. For SDI-9, the maximum drought events are 3 to 4 events for Peninsular Malaysia. Meanwhile, SDI-12 shows the maximum drought is 5 to 6 events in the Northwest and Southwest.

In the 2000s, for SDI-3 and SDI-6, the spatial distributions of drought events are mainly shown in the Northeast and Southeast regions, with 7 to 10 maximum drought events. However, for SDI-9 and SDI-12, the spatial distribution shows the





same pattern, but the number of drought events is slightly lower than 5 to 6. The same patterns were found for SDI-6 and SDI-9. However, the maximum drought events are 5 to 6 for SDI-9. For SDI-12, the distribution pattern of Southeast

distributed with the number of droughts is 5 to 6 events. Figure 5 shows that the number of drought events increased steadily from the 1980s to the 2010s, signifying increased drought frequency over Peninsular Malaysia's entire river basin. In the 2010s, based on SDI-3, Peninsular Malaysia experienced hydrological drought up to 10 times. SDI indicates a total of 2559 hydrological drought events were recorded for the 42 stations, which are 922 drought events for three months (SDI-3), 652 drought events for six months (SDI-6), 545 events for nine months (SDI-9) and 440 drought events for annual (SDI-12). In

time scale (3 to 12 months), the hydrological drought events mainly distributed in the Northeast and Southeast and the few areas in Northwest and Southwest of Peninsular Malaysia.

Figure 5 shows the spatial and temporal variation of SDI values calculated for long periods, representing the 9 and 12 months. The hydrological drought for events with 9 and 12 months durations principally occurred during the 2010s. These

were primarily located in the East Coast and Northern the Southern region of West Coast Peninsular Malaysia. Drought events have changed dramatically in most regions of the river basin between 1980 and 2010. The South-Eastern region is more prone to droughts than any other region.

Meanwhile, most hydrological droughts during the late 21$^{st}$ century occurred in the Southern and Northern region of

Peninsular Malaysia. Furthermore, the short term drought events (a drought less than six months) were distributed throughout Peninsular Malaysia. Interannual variability in short-term drought situations is rather significant.

**3.4 Hydrological drought frequency**

To identify drought-prone regions in Peninsular Malaysia, drought occurrences were analysed on a three-month, six-month,

nine-month, and twelve-month timescale. The frequency of drought conditions classified as moderate, severe, or extreme is evaluated for each station. The purpose of this analysis is to determine which regions are commonly affected by droughts over comparable periods based on their occurrence percentages.

Drought frequency in Peninsular Malaysia is evaluated at different time scales using the percentage occurrence of each event

at each station in relation to the total number of observations in the same category and period scale throughout the region. Figure 6 depicts the spatial distribution characteristics of different drought frequencies during the last 40 years at multiple periods. As shown in Figure 6, the frequency values of four drought categories (in percentage terms) are classified into five frequencies: 0 – 20%, 20 – 40%, 40 – 60%, 60 – 80%, and 80 – 100%. The percentage is provided by taking the ratio of drought occurrences in each time scale to the entire duration of the data record for the same time scale and drought category





across the region. For example, for 1980-1989, the number of monthly SDI-3, which is less than -1 (considered moderate
drought) divided by 120 months.

**Figure 6: Spatial and temporal distribution of hydrological drought frequency (a) three and six months and (b) nine
and twelve months in Peninsular Malaysia**


Table 3 show the percentage of drought occurrences based on the drought frequency categories. Drought occurrence for SDI-
3 was 85.72% and 14.28% for Peninsular Malaysia in the 1980s, and drought occurrence for SDI-6 was 78.57% and 21.43%,
rare and less frequent, respectively. While, for SDI-9, the areas with rare and less frequent accounted for 83.33% and 9.52%
of drought occurrence, respectively, those with often occurrence for 7.14% based on drought occurrences. The hydrological
droughts that occur rarely and less frequently accounted for 61.91% and 26.19% of drought occurrence in the entire basins,
respectively, for SDI-12, while those that occur often accounted for 11.90% of drought occurrence. Drought occurrences for
SDI-3 were 35.71% and 47.62% in the 1990s, respectively, with rare and less occurrences. Droughts often occur in various
parts of Peninsular Malaysia, with 16.67% of the time.  In the time scales of 9 months (SDI-9), no frequent hydrological
drought in the entire basin. Drought occurrences were about 33.33%, 54.76%, and 11.90% for rare, less, and often,
respectively. However, SDI-12 showed an increased frequency of often occurrences, with 23.81% of drought occurrences,
compared to 30.95% and 45.24% for rare and less occurrences of hydrological drought, respectively.

The rare and less frequent occurrences of SDI-3 in the 2000s accounted for 14.28% and 85.71% of drought occurrences in
the entire basin, respectively. However, rare occurrences doubled to 30.95%, whereas drought occurrences declined to
69.04% over six months (SDI-6). The often occurrence of SDI-9 and SDI-12 accounted for 11.90% and 21.43% of drought
occurrence, respectively, whereas those with less occurrence for 59.52% and 52.38% of drought occurrence, and those with
rare occurrence for 28.57% and 26.19% of drought occurrence, respectively. The rare, less, often, and frequent occurrences
of hydrological drought in the 2010s were 38.10%, 52.38%, 7.14%, and 2.38%, respectively, for a time scale of three months
(SDI-3). Above the level of often occurrence demonstrates a significant shift (23.81% and 4.76%, respectively) in the nine-
month (SDI-9) scale, whereas rare and less occurrences accounted for 23.81% and 47.62% drought occurrence, respectively.
The often occurrence of drought did not differ considerably between SDI-12 and SDI-9 (23.81% and 4.76%, respectively),
although the rare and less occurrences were the same (35.72%).

**Table 3 Percentage of Drought Occurrence with Different Time Scales**


Overall, in the time scales of 3 to 12 months, the areas with frequent occurrences tended to increase, with those increasing
significantly in the 1990s and 2010s. However, throughout the 1980s and 2000s, the frequent occurrence of drought did not
happen. In the 1990s and 2010s, the frequency of hydrological droughts was the highest. The hydrological drought



occurrences for frequently and extremely frequently in a particular area are 25.30% and 24.60%, respectively. Based on the 3
and 6 months' time scales (SDI-3 and SDI-6), areas are rare and less frequent to be hit by droughts in the 1980s and 2000s.
For the 1990s and 2010s of SDI-3 and SDI-6, there are often frequent droughts in the Northern and central region of West
Coast Peninsular Malaysia. Considering the hydrological drought category based on the 12 months' time scale, regions where
droughts occurred extremely frequent at two stations (S13 and S28, Selangor and Perak, respectively).

The spatial analysis of drought characteristics also revealed that the majority of areas in Peninsular Malaysia are prone to
short-term droughts, with a relatively high frequency in the Northeast and Southeast, particularly in the central and southern
regions, where the frequency reached 35.7% and 42.8%, respectively. The temporal analysis on SDI-3 indicates an
advantage in recognising hydrological drought in a short-term change, especially by human factors. In other words, since
drought is a natural hazard caused by various factors, SDI-3 appears to be an appropriate index because the streamflow used
considers the effects of each drought-causing factor and affects water supply for domestic or irrigation purposes. In
summary, the results demonstrate that the frequency of droughts decreases as the extent of the drought increases. The result
indicated that for a particular time scale, moderate, severe, and extreme droughts occur most frequently, as predicted during
the past decade. It was evident that the hydrological drought takes up a significant proportion of the historical droughts, and
its covering areas are across most of the basin with the less frequent (20-40%). It can be noted that the percentage occurrence
of drought events of a given category varies with location and SDI time scale. SDI-12 is more appropriate for water resource
management applications. The spatial and temporal distribution in Figure 6 showing the complexity of hydrological drought
with the individual drought episodes can have very different spatial patterns in terms of onset, intensity, spatial propagation,
and area.

**4 Conclusion**

The Streamflow Drought Index was used to evaluate hydrological droughts in Peninsular Malaysia from 1978 to 2018. Using
the SDI, the hydrological drought analysis found that nearly all stations suffered hydrological droughts throughout the study
period. Therefore, calculating the SDI from the different time scales in this paper can reflect the hydrological drought's
spatial and temporal distribution characteristics in Peninsular Malaysia. Extreme drought events mainly occurred in the last
two decades, from 1997 to 1998 to 2001 until 2002, with 2011 to 2018 being the driest years.

1997-1999, 2002, and 2016-2018 are the basin's most critical drought years, with more than 48% of the basin's total area
experiencing hydrological drought. SDI indicates a total of 2559 hydrological drought events were recorded in 40 years
study period, which is 36.03% of drought events for three months (SDI-3), 25.48% drought events for six months (SDI-6),
21.30% events for nine months (SDI-9) and 17.19% drought events for annual (SDI-12). On a time scale (3 to 12 months),





the hydrological drought events mainly distributed in the Northeast and Southeast and few areas in the Northwest and Southwest of Peninsular Malaysia. The highest percentage of drought occurrences is mostly are rare and less occurrences of hydrological drought frequency in Peninsular Malaysia. Most of the areas are prone to relatively high frequency in Northeast and Southeast of Peninsular Malaysia, especially in the central and southern regions, where the frequency reached 35.7% and 450 42.8%, respectively.

The following findings have been obtained:

(1) From the 1980s to the 2010s, Peninsular Malaysia's hydrological drought intensity tended to gradual relief in the 1980s to 1990s and increasing aggravation in the 2000s to 2010s. Furthermore, the 1990s and 2010s had the most severe hydrological 455 droughts, followed by the 1980s and 2000s. Drought extended as time scales increased, whereas regions with severe drought decreased and shifted toward Peninsular Malaysia's centre.

(2) From the 1980s to the 2010s, the number of drought events gradually grew, implying increased drought frequency over Peninsular Malaysia's entire river basin. In time scale (3 to 12 months), the hydrological drought events mainly distributed in the Northeast and Southeast and the few areas in Northwest and Southwest of Peninsular Malaysia.

(3) The frequency of occurrence increased dramatically between the 1990s and 2010s on time scales of SDI-3, SDI-6, SDI-9, and SDI-12. This, however, did not occur between the 1980s and the 2000s. The spatial-temporal distribution of hydrological drought frequency in Peninsular Malaysia corresponds to the severity of the hydrological drought. In the 1990s and 2010s, the hydrological drought frequency was the highest, with frequent occurrence during the interval.

This study highlights the need for selecting a proper time scale in drought assessment. Furthermore, the average duration of a drought varied due to the time scale selected for analysis. Droughts are identified with longer durations at a more prolonged scale, whereas droughts sustained for a shorter period at a shorter scale. The most extended timescale (12 months) identified more extreme drought events than the shortest timescale (3 months). The shortest scale (3 months) was able to detect more mild and moderate events. As the most extended timescale (12 months) is more likely to include dry and wet periods, its 470 high value may produce misleading information to the early warning system. Therefore, the selection of an appropriate timescale is essential for developing an effective drought mitigation strategy. Among SDI timescales, the SDI-3 is most suited for effectively tracking hydrological drought. For tropical regions, it is the most sensitive scale to alterations in streamflow.

**Author contribution**

Hasrul Hazman Hasan: Conceptualisation, writing-original draft, conceptualisation, methodology, formal analysis, validation, investigation. Siti Fatin Mohd Razali: Funding acquisition, supervision, conceptualisation, writing-review and



editing, review of analysis, validation. Nur Shazwani Muhammad: Supervision, conceptualisation, writing-review and editing, review of analysis, validation. Asmadi Ahmad: writing-review and editing. Hasrul Hazman Hasan prepared the manuscript with contributions from all co-authors.

## Competing interests

The authors declare that they have no conflict of interest.

## Acknowledgements

The authors are thankful to the Ministry of Education Malaysia for the financial support of this research through research grant number FRGS/1/2018/TK01/UKM/02/2. The authors would also like to acknowledge their gratitude to the Department of Irrigation and Drainage Malaysia, for providing data. We would like to acknowledge the Ministry of Education (MOE) Malaysia and Universiti Kebangsaan Malaysia (UKM) for supporting this research to be completed successfully.

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





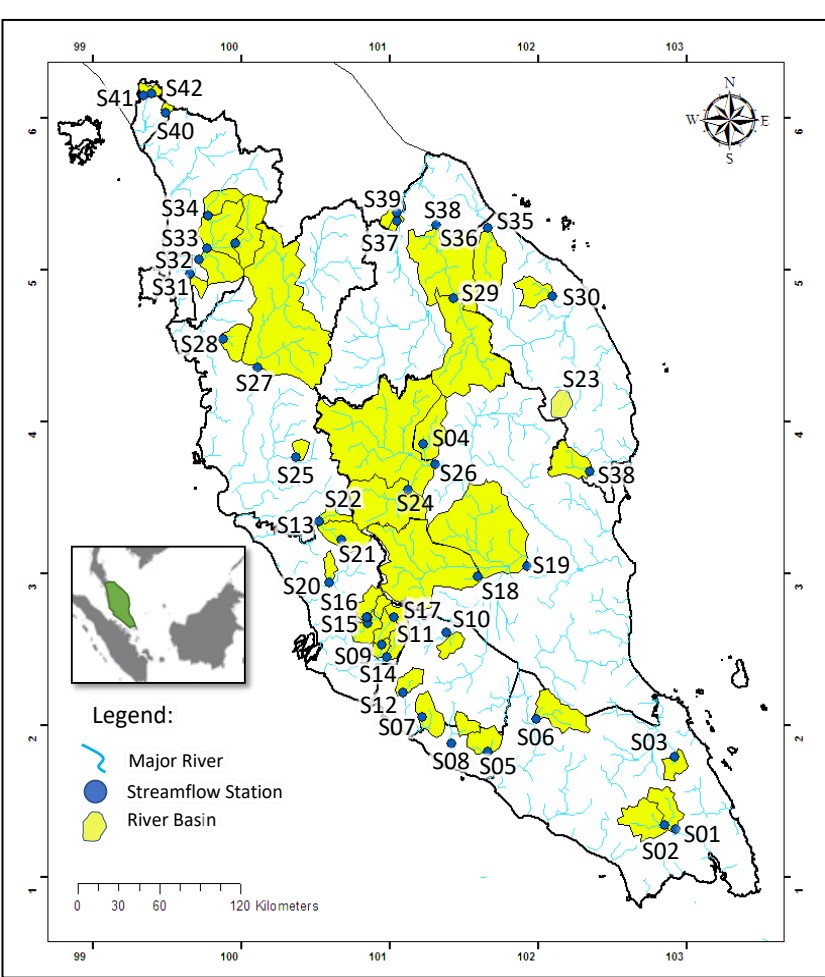

**Figure 1: The location of streamflow stations in Peninsular Malaysia.**






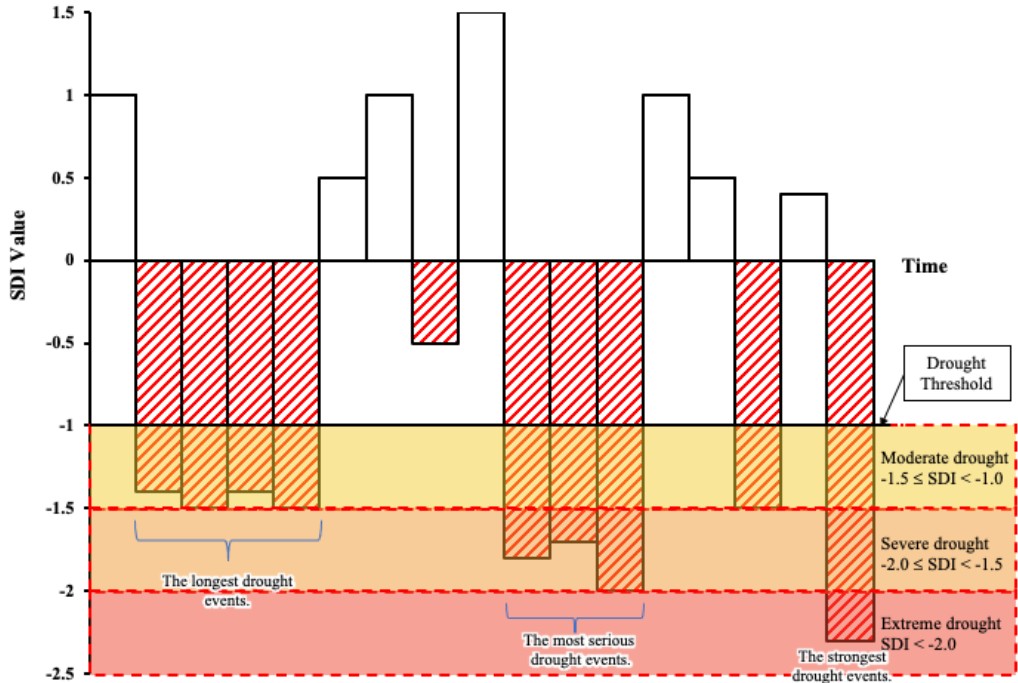

**Figure 2: The three components of a drought event for drought characteristics.**
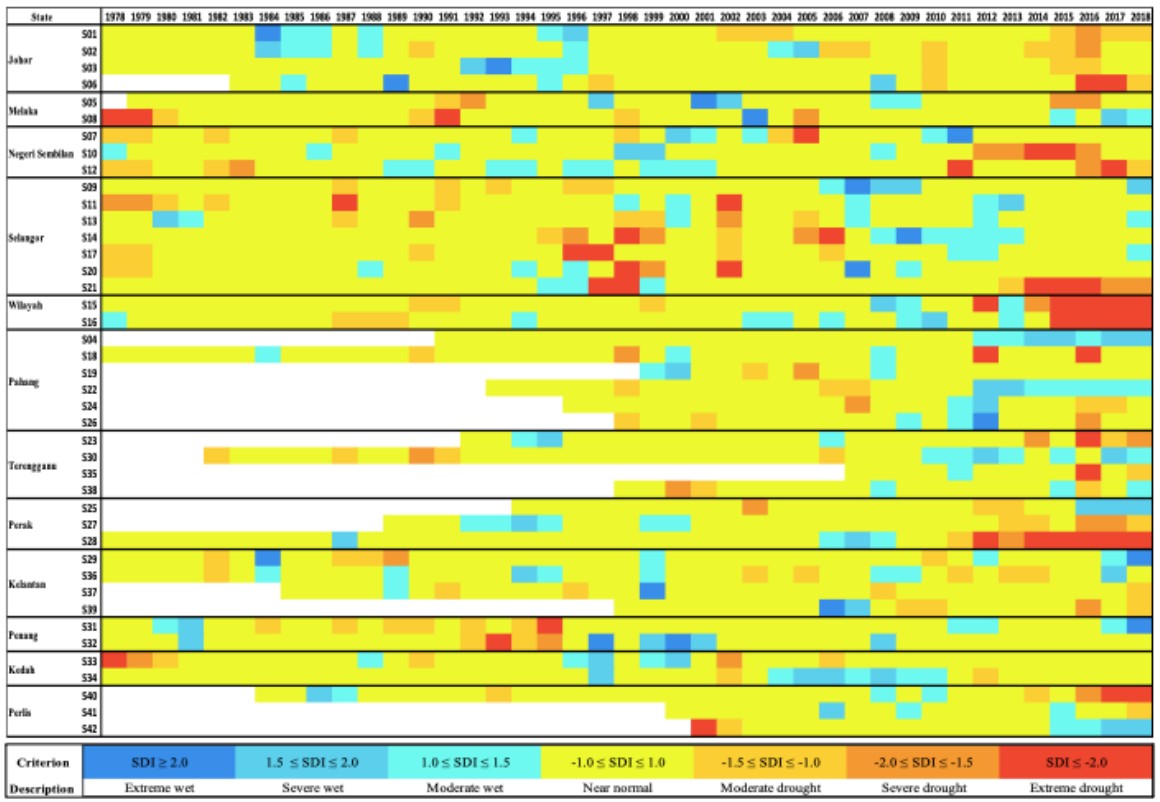

**Figure 3: Colour-coded table of SDI-12 for 40-year time series.**




**(a)**

| Criterion | SDI ≥ 2.0 | 1.5 ≤ SDI < 2.0 | 1.0 ≤ SDI < 1.5 | -1.0 ≤ SDI < 1.0 | -1.5 ≤ SDI < -1.0 | -2.0 ≤ SDI < -1.5 | SDI < -2.0 |
|---|---|---|---|---|---|---|---|
| Description | Extreme wet | Severe wet | Moderate wet | Near normal | Moderate drought | Severe drought | Extreme drought |



**(b)**



| Criterion | SDI ≥ 2.0 | 1.5 ≤ SDI < 2.0 | 1.0 ≤ SDI < 1.5 | -1.0 ≤ SDI < 1.0 | -1.5 ≤ SDI < -1.0 | -2.0 ≤ SDI < -1.5 | SDI < -2.0 |
|---|---|---|---|---|---|---|---|
| Description | Extreme wet | Severe wet | Moderate wet | Near normal | Moderate drought | Severe drought | Extreme drought |



**Figure 4: Spatial and temporal distribution of SDI (a) three and six months and (b) nine and twelve months for ten years interval.**



**(a)**



**(b)**





**Figure 5: Spatial and temporal of hydrological drought events (a) three and six months and (b) nine and twelve months in Peninsular Malaysia.**



**(a)**



**(b)**

| Class | 0% - 20% | 20% - 40% | 40% - 60% | 60% - 80% | 80% - 100% |
|---|---|---|---|---|---|
| Description | Rare | Less | Often | Frequent | Extreme frequent |





**Figure 6: Spatial and temporal distribution of hydrological drought frequency (a) three and six months and (b) nine and twelve months in Peninsular Malaysia.**



**Table 1. Information on Streamflow Stations in Peninsular Malaysia**

| Station ID | Gauging Station | River | State | Area (km²) | Coordinate | | Record Duration (Year) | Start and End Year |
|---|---|---|---|---|---|---|---|---|
| | | | | | Latitude (N) | Longitude (E) | | |
| S01 | 1737451 | Johor River | Johor | 1130 | 1° 46' 50.002" N | 103° 44' 44.999" E | 40 | 1978-2018 |
| S02 | 1836402 | Sayong River | Johor | 624 | 1° 48' 15.001" N | 103° 40' 09.998" E | 40 | 1978-2018 |
| S03 | 2237471 | Lenggor River | Johor | 207 | 2° 15' 29.999" N | 103° 44' 10.000" E | 40 | 1978-2018 |
| S04 | 4320401 | Kecau River | Pahang | 497 | 2° 17' 25.001" N | 102° 29' 35.002" E | 28 | 1990-2018 |
| S05 | 2224432 | Kesang River | Melaka | 161 | 2° 20' 35.002" N | 102° 15' 10.001" E | 40 | 1978-2018 |
| S06 | 2528414 | Segamat River | Johor | 658 | 2° 30' 24.998" N | 102° 49' 05.002" E | 25 | 1993-2018 |
| S07 | 2519421 | Linggi River | Negeri Sembilan | 523 | 2° 31' 00.001" N | 102° 03' 29.999" E | 40 | 1978-2018 |
| S08 | 2322413 | Melaka River | Melaka | 350 | 2° 40' 52.000" N | 101° 55' 39.000" E | 40 | 1978-2018 |
| S09 | 2917401 | Langat River | Selangor | 380 | 2° 54' 55.001" N | 101° 49' 25.000" E | 40 | 1978-2018 |
| S10 | 3022431 | Triang River | Negeri Sembilan | 904 | 2° 59' 34.001" N | 101° 47' 12.998" E | 40 | 1978-2018 |
| S11 | 2816441 | Langat River | Selangor | 1240 | 2° 59' 39.998" N | 101° 47' 10.000" E | 40 | 1978-2018 |
| S12 | 2920432 | Triang River | Negeri Sembilan | 228 | 3° 04' 30.000" N | 102° 13' 05.002" E | 40 | 1978-2018 |
| S13 | 3813411 | Bernam River | Selangor | 1090 | 3° 08' 20.000" N | 101° 41' 49.999" E | 40 | 1978-2018 |
| S14 | 2918401 | Semenyih River | Selangor | 225 | 3° 10' 25.000" N | 101° 52' 19.999" E | 40 | 1978-2018 |



| Station ID | Gauging Station | River | State | Area (km²) | Coordinate | | Record Duration (Year) | Start and End Year |
|---|---|---|---|---|---|---|---|---|
| | | | | | Latitude (N) | Longitude (E) | | |
| S15 | 3116433 | Gombak River | Wilayah Persekutuan | 122 | 3° 10' 25.000" N | 101° 41' 49.999" E | 40 | 1978-2018 |
| S16 | 3116434 | Batu River | Wilayah Persekutuan | 145 | 3° 10' 35.000" N | 101° 41' 15.000" E | 40 | 1978-2018 |
| S17 | 3118445 | Lui River | Selangor | 68 | 3° 24' 10.001" N | 101° 26' 35.002" E | 40 | 1978-2018 |
| S18 | 3424411 | Pahang River | Pahang | 19000 | 3° 26' 39.998" N | 102° 25' 45.001" E | 40 | 1978-2018 |
| S19 | 3527410 | Pahang River | Pahang | 25600 | 3° 30' 45.000" N | 102° 45' 29.999" E | 20 | 1998-2018 |
| S20 | 3414421 | Selangor River | Selangor | 1450 | 3° 41' 07.001" N | 101° 31' 23.999" E | 40 | 1978-2018 |
| S21 | 3615412 | Bernam River | Selangor | 186 | 3° 48' 27.000" N | 101° 22' 09.998" E | 40 | 1978-2018 |
| S22 | 4019462 | Lipis River | Pahang | 1670 | 4° 01' 05.002" N | 101° 57' 55.001" E | 26 | 1992-2018 |
| S23 | 4930401 | Berang River | Terengganu | 140 | 4° 08' 00.000" N | 103° 10' 30.000" E | 27 | 1991-2018 |
| S24 | 4121413 | Jelai River | Pahang | 7320 | 4° 11' 10.000" N | 102° 08' 39.998" E | 23 | 1995-2018 |
| S25 | 4212467 | Chendering River | Perak | 119 | 4° 13' 54.998" N | 101° 13' 09.998" E | 25 | 1993-2018 |
| S26 | 4223450 | Tembeling River | Pahang | 5050 | 4° 19' 14.999" N | 102° 03' 40.000" E | 21 | 1997-2018 |
| S27 | 4809443 | Perak River | Perak | 7770 | 4° 49' 09.998" N | 100° 57' 55.001" E | 30 | 1988-2018 |
| S28 | 5007421 | Kurau River | Perak | 337 | 5° 00' 45.000" N | 100° 43' 54.998" E | 40 | 1978-2018 |
| S29 | 5222452 | Lebir River | Kelantan | 2430 | 5° 16' 30.000" N | 102° 16' 00.001" E | 40 | 1978-2018 |
| S30 | 5229436 | Nerus River | Terengganu | 393 | 5° 17' 30.001" N | 102° 55' 19.999" E | 37 | 1981-2018 |
| S31 | 5405421 | Kulim | Penang | 129 | 5° 26' 10.000" N | 100° 30' 50.000" E | 40 | 1978-2018 |




| Station ID | Gauging Station | River | State | Area (km²) | Coordinate | | Record Duration (Year) | Start and End Year |
|---|---|---|---|---|---|---|---|---|
| | | | | | Latitude (N) | Longitude (E) | | |
| | | River | | | | | | |
| S32 | 5505412 | Muda River | Penang | 4010 | 5° 31' 54.998" N | 100° 34' 19.999" E | 40 | 1978-2018 |
| S33 | 5606410 | Muda River | Kedah | 3330 | 5° 36' 34.999" N | 100° 37' 35.000" E | 40 | 1978-2018 |
| S34 | 5806414 | Muda River | Kedah | 1710 | 5° 38' 20.000" N | 100° 48' 45.000" E | 40 | 1978-2018 |
| S35 | 5724411 | Besut River | Terengganu | 787 | 5° 44' 20.000" N | 102° 29' 35.002" E | 12 | 2006-2018 |
| S36 | 5721442 | Kelantan River | Kelantan | 11900 | 5° 45' 45.000" N | 102° 09' 00.000" E | 40 | 1978-2018 |
| S37 | 5718401 | Golok River | Kelantan | 80 | 5° 47' 10.000" N | 101° 53' 30.001" E | 34 | 1984-2018 |
| S38 | 4131453 | Cherul River | Terengganu | 505 | 5° 49' 09.998" N | 100° 37' 54.998" E | 21 | 1997-2018 |
| S39 | 5818401 | Lanas River | Kelantan | 216 | 5° 50' 25.001" N | 101° 53' 30.001" E | 21 | 1997-2018 |
| S40 | 6503401 | Arau River | Perlis | 21 | 6° 30' 10.001" N | 100° 21' 05.000" E | 35 | 1983-2018 |
| S41 | 6602402 | Pelarit River | Perlis | 21 | 6° 36' 45.000" N | 100° 12' 24.998" E | 19 | 1999-2018 |
| S42 | 6602403 | Jarum River | Perlis | 22 | 6° 37' 30.000" N | 100° 15' 37.001" E | 18 | 2000-2018 |




**Table 2 Hydrological Drought Classification by SDI**

| SDI values | Classification |
|---|---|
| SDI ≥ 2.0 | Extreme wet |
| 1.5 ≤ SDI < 2.0 | Severe wet |
| 1.0 ≤ SDI < 1.5 | Moderate wet |
| -1.0 ≤ SDI < 1.0 | Near normal |
| -1.5 ≤ SDI < -1.0 | Moderate drought |
| -2.0 ≤ SDI < -1.5 | Severe drought |
| SDI < -2.0 | Extreme drought |

**Table 3. Percentage of Drought Occurrence with Different Time Scales**

| Year | Drought Frequency Category | Drought Occurrences (%) | | | |
|---|---|---|---|---|---|
| | | SDI-3 | SDI-6 | SDI-9 | SDI-12 |
| 1980s | Rare | 85.72 | 78.57 | 83.34 | 61.91 |
| | Less | 14.28 | 21.43 | 9.52 | 26.19 |
| | Often | - | - | 7.14 | 11.90 |
| | Frequent | - | - | - | - |
| | Extremely frequent | - | - | - | - |
| 1990s | Rare | 35.71 | 35.71 | 33.34 | 30.95 |
| | Less | 47.62 | 40.58 | 54.76 | 45.24 |
| | Often | 16.67 | 21.32 | 11.90 | 23.81 |
| | Frequent | - | 2.39 | - | - |
| | Extremely frequent | - | - | - | - |
| 2000s | Rare | 14.28 | 30.95 | 28.57 | 26.19 |
| | Less | 85.72 | 69.05 | 59.53 | 52.38 |
| | Often | - | - | 11.90 | 21.43 |
| | Frequent | - | - | - | - |
| | Extremely frequent | - | - | - | - |
| 2010s | Rare | 38.10 | 38.09 | 23.81 | 35.71 |
| | Less | 52.38 | 47.20 | 47.62 | 35.72 |
| | Often | 9.52 | 11.90 | 23.81 | 23.81 |
| | Frequent | - | 2.81 | 4.76 | 4.76 |
| | Extremely frequent | - | - | - | - |