# Peer review of "Hydrological Drought across Peninsular Malaysia: Implication of drought index"

_Natural Hazards and Earth System Sciences, 2021_

## Author Comment (AC2)

**Comment on nhess-2021-176**
**Anonymous Referee #2**
Referee comment on "Hydrological Drought across Peninsular Malaysia: Implication of drought index" by Hasrul Hazman Hasan et al., Nat. Hazards Earth Syst. Sci. Discuss., https://doi.org/10.5194/nhess-2021-176-RC2, 2021

**General comment**
The manuscript illustrates the application of a standard hydrological drought index (Streamflow Drought Index, SDI) for the detection at the regional scale of drought events; the case study is the peninsular Malaysia. This work contributes to the state of the art on the topic by improving the knowledge about the hydrology of the case study region. The topic is of interest for the hydrologic community, yet the manuscript needs additional efforts from the Authors to clarify some aspects that are fundamental for the reader understanding; further, a deeper analysis based on the available data is expected. Specific comments follow.
**Response:** We are grateful to the reviewer for their time and suggestions in helping to improve the manuscript.

**Specific comments**
From the abstract alone it is not clear which is the content and main objective of this work; further, it appears that this work is simply a case study application of methods already known in the literature. If this is true, it should be emphasized the innovative contribution provided by this work.
**Response:** We thank you for these comments. We have revised the summary based on your recommendation on page 1, lines 11-17 and lines 24-26. We have rewritten the abstract to better understand the research objectives and edited it so that the methods are reflected in the results and the data support the conclusions.

Also the Introduction Section needs additional efforts from the Authors to better state the research gaps that justify the proposed work and to avoid repetitions. I'm not sure that the literature review covers properly what has been already proposed in the literature in terms of drought indexes development and application. Further, while the Authors states that there are not many work on SDI (l. 108-109), they report a non-negligible number of reference on its application at several time-scales (l. 151-156).
**Response:** Thank you very much for your suggestion. We agree with the reviewer. We have revised the manuscript based on your recommendation. Please refer to page 3 (lines 91-96) and page 4 lines 99 to 101, 106-120, 127-128, 133-134.

It is not clear which is the motivation for the choice of the period within the year where SDI is computed, starting in January and covering 3, 6, 9 and 12 months (l. 165-67). In other words, its should be explained why, e.g., the 9-months SDI refers to the period from January to September and not to another one (e.g. from April to December).
**Response:** There were no four seasons in Peninsular Malaysia which is tropical country. Therefore, we can define the hydrological year as beginning each year with January. This corresponds to the tropical climate of Peninsular Malaysia with various monsoons and monsoon transitions. A detailed analysis of the definition of the hydrological year can be found in Hirsch and Fisher (2014).

SDI is an index that allows to detect drought events when it is below a given threshold value. To compute drought frequency it should be first defined a drought event; hence, section 2.3 should follow 2.4.

**Response:** We agree with the reviewer. Therefore, section 2.4 has been included in section 2.3 Identification of drought events (page 7).

Why depicting results averaged over 10-years time windows? Which is the difference between depicting the number of droughts and the frequency of drought events? Which is the statistical significance of frequency computed over a short time period of 10 years? I personally believe that the presentation of results should be improved and more details should be added.

**Response:** Thank you for your comments. The use of a time interval of 10 years or decades is due to the fact that many equations for hydrological modelling or projection use decades to overcome the degree of bias in the models towards underestimation of streamflow (Daniels, 2014). Besides, this is one of the objectives of the present work. We have improved the presentation of the results in Table 3 and page 13 (lines 399-408 and 417-426), page 14 (lines 419-420 and 434-437). The reviewer may also refer to Figures 3-6.

Drought events can be quantified (as indicated in the Methodology Section) in terms of three different quantities, intensity, duration and severity. How do those quantities are used here to understand drought phenomenon over Malaysia? I personally believe that available data are not exploited enough for drought understanding.

**Response:** Thank you very much for your suggestion. This study focused on the hydrological characteristics of droughts in terms of the number of drought events and the frequency of occurrence of droughts at different time scales.

Results are presented in detail yet not discussed in terms of possible physical explanation of the observed phenomenon. Hydrological droughts result from different processes, as clearly mentioned in the introduction section; yet, there is not reference to such processes.

**Response:** Thank you for your suggestion. We have improved the results sections to meet the objectives of the study.

**Reference**

Daniels, B. J.: Effects of climate nonstationarity on low-flow models for southern New England. [online] Available from: http://160.75.22.2/docview/1613245669?accountid=11638%5Cnhttp://re5tm7xf6s.search.serialssolutions.com/?ctx_ver=Z39.88-2004&ctx_enc=info:ofi/enc:UTF-8&rfr_id=info:sid/ProQuest+Dissertations+%26+Theses+Global&rft_val_fmt=info:ofi/fmt:kev:mtx:dissertation&r, 2014.

Hirsch, R. M. and Fisher, G. T.: Past, Present, and Future of Water Data Delivery from the U.S. Geological Survey, J. Contemp. Water Res. Educ., 153(1), 4–15, doi:10.1111/j.1936-704x.2014.03175.x, 2014.

---

## Author Comment (AC3)

**Comment on nhess-2021-176**
**Anonymous Referee #3**
Referee comment on "Hydrological Drought across Peninsular Malaysia: Implication of drought index" by Hasrul Hazman Hasan et al., Nat. Hazards Earth Syst. Sci. Discuss., https://doi.org/10.5194/nhess-2021-176-RC3, 2021.

**General Comments**
The manuscript is investigating hydrological drought in Malaysia employing the Streamflow Drought Index and the run theory, which is of interest to the Journal's audience. However, since both techniques are established, the authors should emphasize the significance and the contribution of their work. In addition, there are technical issues that the authors need to rectify in order for the analysis to be accurate. The language of the manuscript needs some improvement. My evaluation is that the manuscript does not meet NHESS's scientific quality standards and I suggest rejection of the manuscript or conditional acceptance after major revisions.
**Response:** We are grateful to the reviewer for their time and suggestions in helping to improve the manuscript.

**Specific Comments**
In the introduction, the authors need to create a narrative, based on pertinent literature, that explains the contribution of their study to the reader. Lines 71-80 do not contribute towards this goal. Information about SDI in different regions does not need to be included with such detail. On the contrary, the authors need to cite and elaborate on drought studies for Malaysia in order to establish what is the new knowledge that this study is offering.
**Response:** Thank you for your comments. We agree with the reviewer. We have revised the manuscript based on your recommendation in Page 3 (line 75-84) and Page 4 (line 103-120).

Malaysia has a high hydropower potential and several dams (including ones for storage) constructed since the 1960s until recently. The authors also mention at lines 91-93 that seven dams had significantly lower water levels due to drought conditions in 2016. Have the authors performed flow naturalization to remove the effects of upstream flow regulations for the gauges that have a dam upstream? If no, why? For the provided figures it is not clear if there are dams upstream of the gauges. Anthropogenic interventions need to be excluded if the authors intend to evaluate how hydrological drought characteristics have changed throughout the study period. In addition, this comment is critical for the spatial analysis of hydrological drought characteristics across Malaysia.
**Response:** We thank you for your suggestion. We have improved Figure 1 and include the location of the dams in Peninsular Malaysia. However, the naturalisation procedure of the streamflow indices is essentially intended for the regionalisation analysis of hydrological drought. This method is not addressed in this study. We focus on historical hydrological drought in individual watersheds and determine the relationship between hydrological drought and the spatial-temporal analysis using data collected from a specific river in the catchment of interest. The relationship between spatial-temporal was represented in Figure 4-6.

Based on run theory and the authors' definition of drought characteristics, drought severity is equal to the shaded area below the threshold --- here set to −1 --- (Yevjevich 1967), not the shaded area below the horizontal axis. Equation 4 should reflect that. The analysis needs to be redone.

**Response:** Thank you very much for your comments. Once the SDI is calculated, specific criteria must be used to identify drought events. In this study, the ToR originally proposed by Yevjevich (1967) was applied to determine hydrological drought characteristics (Razmkhah, 2017). Following Nalbantis and Tsakiris (2009), the successive sequence of months with SDI values ($X_t$) below the threshold ($X_{-1}$) is defined as a hydrological drought event (Figure 2).

[Figure]

**Figure 2: The determination of hydrological drought characteristics using theory of runs.**

The drought period is the month when SDI values fall below -1.0, indicating the beginning of a drought episode. The drought period is the time between the occurrence of the drought and the time of its end. The cumulative drought index defines the severity of the drought during a drought event. The onset of the drought was determined at the beginning of the period when the SDI was negative for an extended period. The end of the drought was expected for the first month in which the SDI became positive.

The authors at Line 135 state that they include in their analysis 42 gauge stations with 40 years of continuous streamflow data. In line 141 the authors state that 17 of those have a record of less than 40 years. However, Table 2 indicates that there are 12 stations with less than 30 years of record, with the smallest time series being just 12 years. It is recommended to have a record of 30 years or more to accurately compute a standardized drought index (e.g. SPI, SDI, etc.). Nalbantis & Tsakiris (2009) used a record of 30 years. The gauges with data less than 30 years need to be dropped from the analysis.

**Response**: Thank you for your comments. Some studies use streamflow record data that are less than 30 years old. For example, Yeh et al. (2015) used 28 years of streamflow record, Sardou and Bahremand, (2014) used 25 years of record data, and Sohoulande Djebou, (2019) used 19 years of streamflow data to derive SDI values. Furthermore, this study uses the 10-

year interval method for spatial and temporal analysis. Therefore, stations less than 30 years are not removed for IDW analysis due to drought variability between 10-year intervals.

I second the comment of Anonymous Referee #2 about what the results section is missing.
**Response:** Thank you for your suggestion. We have improved the results sections to meet the objectives of the study.

**References**

Nalbantis, I. and Tsakiris, G.: Assessment of hydrological drought revisited, Water Resour. Manag., 23(5), 881–897, doi:10.1007/s11269-008-9305-1, 2009.

Razmkhah, H.: Comparing Threshold Level Methods in Development of Stream Flow Drought Severity-Duration-Frequency Curves, Water Resour. Manag., 31(13), 4045–4061, doi:10.1007/s11269-017-1587-8, 2017.

Sardou, F. S. and Bahremand, A.: Hydrological Drought Analysis Using SDI Index in Halilrud Basin of Iran, Environ. Resour. Res., 2(1), 1, 2014.

Sohoulande Djebou, D. C.: Streamflow Drought Interpreted Using SWAT Model Simulations of Past and Future Hydrologic Scenarios: Application to Neches and Trinity River Basins, Texas, J. Hydrol. Eng., 24(9), 05019024, doi:10.1061/(asce)he.1943-5584.0001827, 2019.

Yeh, C. F., Wang, J., Yeh, H. F. and Lee, C. H.: SDI and Markov chains for regional drought characteristics, Sustain., 7(8), 10789–10808, doi:10.3390/su70810789, 2015.

Yevjevich, V.: An objective approach to definitions and investigations of continental hydrologic drought, in Hydrology Paper, vol. 23., 1967.

---

## Author Comment (AC4)

**Comment on nhess-2021-176**
**Anonymous Referee #4**

Referee comment on "Hydrological Drought across Peninsular Malaysia: Implication of drought index" by Hasrul Hazman Hasan et al., Nat. Hazards Earth Syst. Sci. Discuss., https://doi.org/10.5194/nhess-2021-176-RC4, 2021

This study aims at investigating the spatial and temporal variations of hydrological drought in Peninsular Malaysia for the period 1978-2018 by Streamflow Drought Index (SDI) using streamflow data recorded at 42 stations. The drought was also characterized at four time scales of 3-, 6-, 9- and 12-month.
**Response:** We thank the reviewers for their time and suggestions, which helped to improve the manuscript.

1- My main concern is on the novelty of this work, especially when it was submitted to the special issue "Recent advances in drought and water scarcity monitoring, modelling, and forecasting". Hydrological drought was characterized by Streamflow Drought Index (SDI) as developed by Nalbantis & Tsakiris (2009) without any modification. Drought characteristics were identified by the run theory (Yevjevich 1967) and the interpolation was done by the well-known Inverse Distance Weighting (IDW) method.
**Response:** Thank you for your comments. We have revised the manuscript based on your recommendation. Please refer to page 4, lines 103 to 128.

2- L147-149: " The main advantage of SDI is that it requires fewer data than other indices, such as the Palmer Hydrological Drought Index, which need streamflow and rainfall data. The selection of SDI is because of the availability of streamflow data." Does it mean rainfall data are not available in Peninsular Malaysia? In addition, as mentioned in lines 64-66 of the manuscript "several indices are using only streamflow data, namely, Regional Streamflow Deficiency Index (RSDI), Standardized Streamflow Index (SSFI), Streamflow Drought Index (SDI), Baseflow Index (BFI) and Regional Drought Area Index (RDAI)". Why was the SDI used here?
**Response:** Thank you very much for your comment. This study is about determining hydrological drought using streamflow data. In Malaysia, many studies have been conducted on meteorological drought based on rainfall data, e.g. Ahmad and Deni, (2013); Hong and Hong, (2016); Sanusi et al. (2015). So far, only a few studies have been conducted on hydrological drought based on SDI. There is only one study related to the application of SDI, which is the study by Khan et al. (2017). However, this study concentrates on the state of Selangor. In contrast, this paper focused on hydrological drought for all available hydrological stations at Peninsular Malaysia. To our knowledge, this is the first comprehensive study to examine the multi-time scales of observed streamflow at 42 stations on Peninsular Malaysia.

L107: "Due to the scarcity of research on hydrological drought monitoring using SDI". In Peninsular Malaysia? Because there are several studies using SDI in other parts of the world that were not cited in the paper. Have all the above indices been used before in Peninsular Malaysia?
**Response:** We thank you for your comments. To our knowledge, this is the first comprehensive study to examine multiple time scales using SDI in observed streamflow at 42 stations in Peninsular Malaysia. We have revised the manuscript based on your recommendation on page 3, lines 75 to 84.

Minor comments:

L82-85: It implies that the El Nino event in the year 1997-1998 was caused by climate change. If so, a reference is needed. If not, revise.

**Response:** Thank you for your suggestions. We have revised the manuscript based on your recommendation on page 3, lines 87-96.

L157: "For a relatively more detailed drought index, the SDI can be computed based on the monthly streamflow value". Most of the drought indices use monthly or smaller-scale data.

**Response:** Thank you for your suggestions. We agreed with the comment. We have revised the manuscript based on your recommendation on page 6, lines 174-176.

L472: "For tropical regions, it is the most sensitive scale to alterations in streamflow." Isn't it the case everywhere because of the smoothing effect at longer scales?

**Response:** Thank you very much for your comments. In this study, the running series of total streamflow volumes for 3, 6, 9 and 12 months were used to derive the SDI series. In this way, minor drought or dependent droughts can be taken into account. The time scale of 12-month simplifies the comparison between the long-term variation of the dry climate and the corresponding hydrological variation.

**References**

Ahmad, N. H. and Deni, S. M.: Homogeneity Test on Daily Rainfall Series for Malaysia, Matematika, 29(1), 141–150 [online] Available from: http://www.matematika.utm.my/index.php/matematika/article/view/586, 2013.
Hong, J. L. and Hong, K. A.: Regional Drought Rainfall for Selangor River Basin in Malaysia Estimated Using L-Moments, Int. J. Hybrid Inf. Technol., 9(6), 413–432, doi:10.14257/ijhit.2016.9.6.37, 2016.
Khan, M. M. H., Muhammad, N. S. and El-shafie, A.: Drought Characterisation in Peninsular Malaysia Using DrinC Software, Pertanika J. Sci. Technol., 25, 81–90, 2017.
Sanusi, W., Jemain, A. A., Zin, W. Z. W. and Zahari, M.: The drought characteristics using the first-order homogeneous Markov Chain of monthly rainfall data in Peninsular Malaysia, Water Resour. Manag., 29(5), 1523–1539, doi:10.1007/s11269-014-0892-8, 2015.

---

## Author Comment (AC5)

**Comment on nhess-2021-176**
**Anonymous Referee #5**
Referee comment on "Hydrological Drought across Peninsular Malaysia: Implication of drought index" by Hasrul Hazman Hasan et al., Nat. Hazards Earth Syst. Sci. Discuss., https://doi.org/10.5194/nhess-2021-176-RC5, 2021

This paper attempts to deal with a very difficult issue which is drought monitoring. The authors used only one index [Streamflow Drought Index (SDI)] for monitoring drought across Peninsular Malaysia.

**Response:** We are grateful to the reviewer for their time and suggestions in helping to improve the manuscript.

I have many concerns regarding the appropriateness of this manuscript for publication in this high-impact journal and especially in the special issue "Recent advances in drought and water scarcity monitoring, modeling, and forecasting". The specific manuscript was presented with no innovative point of view regarding the advantages in the topic of the SI. The contribution of this research in the literature is very weak and unclear. Specifically, the authors used the well-known SDI drought index and simply discussed the results. The paper seems to be more a technical report than a research paper and this can be obvious concerning the structure and the results of this work. Also, the proposed approach seems to have strong local applicability.

**Response:** Thank you for your comments. We have revised the manuscript based on your recommendation in Page 1 (lines 11-17 and 24-27), Page 3 (lines 75-84), Page 4 (lines 99-120 and 125-128) and Page 5 (lines 133-134).

The authors should highlight the contribution of their work in regards to the previously published works. Also, the authors should mention extra information about the existence or not of drought early warning systems across Peninsular Malaysia. What about the flash drought monitoring processes in the study area?

**Response:** Thank you for your comments. We have revised the manuscript based on your recommendation on page 1 (lines 11-17 and 24-27) and page 4 (lines 99-120 and 125-128). The drought monitoring programme in Peninsular Malaysia was launched in early 2001. It was established after the 1998 drought, which affected many residents of Klang Valley. However, drought monitoring in Peninsular Malaysia is based on the percentage deviation from the long-term mean (LTM) of three moving monthly rainfall totals, which serve as indicators of the condition of the catchment. Therefore, this study was conducted to develop a simplified hydrological drought methodology using streamflow data.